# Sublethal Effects of Diamide Insecticides on Development and Flight Performance of *Chloridea virescens* (Lepidoptera: Noctuidae): Implications for Bt Soybean Refuge Area Management

**DOI:** 10.3390/insects11050269

**Published:** 2020-04-28

**Authors:** Lucas Silva Barros, Pedro Takao Yamamoto, Paul Merten, Steve E. Naranjo

**Affiliations:** 1Department of Entomology and Acarology, Luiz de Queiroz College of Agriculture (ESALQ), University of São Paulo (USP), Av. Pádua Dias 11, Piracicaba, São Paulo 13418-900, Brazil; pedro.yamamoto@usp.br; 2USDA-ARS, Arid-Land Agricultural Research Center, 21881 North Cardon Lane, Maricopa, AZ 85138, USA; paul.merten@usda.gov (P.M.); steve.naranjo@usda.gov (S.E.N.)

**Keywords:** tobacco budworm, flubendiamide, chlorantraniliprole, life history, flight mill, Bt soybean

## Abstract

High-dose and refuge are the most important strategies for delaying resistance evolution in Bt crops. Insecticide sprays in refuge areas could be necessary and may limit refuge effectiveness. Here, we evaluated the sublethal effects of two diamide insecticides (chlorantraniliprole and flubendiamide) on *Chloridea virescens* life history traits and flight performance. Sublethal concentrations of chlorantraniliprole and flubendiamide increased larval and pre-pupal development times and decreased larval weight; flubendiamide increased pupal development times. Chlorantraniliprole increased adult male longevity and reduced female fertility, while flubendiamide reduced fecundity. Overall life table parameters were negatively impacted by both treatments. Males exposed to either insecticide showed significant reductions in flight duration and distance for unsustained flights (<30 min). The duration and distance of the first flights were reduced when exposed to chlorantraniliprole. Sustained flights (>30 min) were generally unaffected by insecticide exposure and both sexes flew >6400 m in a single flight. The sublethal effects of flubendiamide and chlorantraniliprole on *C. virescens’* population dynamics could lead to generation asynchrony and provide insufficient susceptible moths when sprayed on refuge crops. However, the distance and duration of flight may still be sufficient to ensure mixing of potentially resistant and susceptible populations from refuge plots.

## 1. Introduction

The tobacco budworm, *Chloridea* (=*Heliothis*) *virescens* Fabricius [Lepidoptera: Noctuidae]) is an important economic pest of several crops in southern Canada, the United States, and throughout South America, except for Chile and southern Argentina [1,2,3]. In Brazil, the pest attacks cotton and the vegetative (leaves and stems) and reproductive structures (flower buds and pods) of soybeans, causing yield loss [4]. In the 2013–2014 growing season, genetically modified soybeans MON 87701 × MON 89788 (Intacta RR2 PRO^®^) were commercially introduced in Brazil. This soybean event expresses genes encoding the insecticidal protein Cry1Ac of *Bacillus thuringiensis* Berliner (Bt) and the 5-enolpyruvylshikimate-3-phosphate synthase (EPSPs) protein of *Agrobacterium* sp. that confers tolerance to the herbicide glyphosate (RR) [5].

Previous work revealed that soybean MON 87701 × MON 89788 provides a “high dose” against major lepidopteran pests such as *C. virescens* [6,7]. The high-dose strategy requires that Bt plants express high enough concentrations of *B. thuringiensis* insecticidal proteins to ensure mortality of more than 95% of heterozygous insects [8,9]. Because of this high efficacy and associated yield protection, soybean MON 87701 × MON 89788 has been widely adopted by growers in most regions of Brazil. As a consequence, there is a high risk for the evolution of resistance in this pest [10,11].

To manage evolution of resistance, the main insecticide resistant management (IRM) strategies are “high dose” and planting of “structured refuge” (non-Bt soybean). The structured refuge areas provide sources of *Bt*-susceptible pests, which can mate with rare survivors from the Bt crop field, decreasing the abundance of the resistant insects [8,9]. Recent global monitoring revealed a sustained susceptibility for populations of nine species of lepidopteran pests from six countries after at least 10 years of exposure to Bt crops [12]. However, in tropical countries such as Brazil, structured refuge adoption and pest management are challenging because of the high pest pressure [4,13,14]. Therefore, although Bt soybean has been highly effective against *C. virescens*, supplemental foliar insecticide applications to control and reduce pest populations could be necessary in non-Bt soybean refuge areas [14,15]. Thus, the Insecticide Resistance Action Committee (IRAC) biotechnology subunit group in Brazil has proposed integrated pest management (IPM) tactics for management of refuge fields, including windows of insecticide sprays when population densities exceed economic thresholds [15].

Diamides (IRAC Group 28) are the most recent chemical group introduced to the insecticide market for use in soybean refuge areas to manage *C. virescens* [16,17,18]. Two representatives from this class of insecticides are flubendiamide, a phthalic acid diamide, and chlorantraniliprole, an anthranilic diamide [19]. Diamide insecticides have a unique mode of action as modulators of ryanodine receptors (RyRs), which are located in the membrane of the sarcoplasmic reticulum of muscle tissues. These channels work to rapidly release Ca^2+^ from intracellular stocks, a process necessary for muscle contraction. In intoxicated insects, symptoms begin with cessation of feeding and uncoordinated muscle contraction, eventually causing mortality [20,21,22,23,24].

In the field, it is likely that in addition to direct mortality (lethal effect), some target pests may be exposed to sublethal concentrations, where they survive but suffer negative biological effects [25,26]. Management of target pests in refuge crops is expected and the size of the refuge crop relative to the Bt crop is adjusted so that sufficient susceptible insects are generated [9,15]. Sublethal effects on target pest biology are not necessarily accounted for in the refuge strategy [27]. Thus, it is important to evaluate and understand how the sublethal effect of any insecticide might affect the overall resistance management strategy [28,29,30,31]. Diamide insecticides can affect muscle contraction and release of neurotransmitters. Thus, it is possible that dispersal and other biological attributes could be negatively affected [29,30,31,32,33,34,35,36,37]. This, in turn, might interfere with the ability of susceptible moths to disperse and mate with resistant moths arising from Bt crop fields and disrupt the high-dose and refuge IRM strategy. We, therefore, used *C. virescens* in Bt soybeans as a model system to study potential effects of sublethal insecticide exposure on aspects of a resistance management system. Our objectives were to evaluate the sublethal effects of flubendiamide and chlorantraniliprole on life history traits of *C. virescens*, life table parameters, and flight performance in order to better understand the implications for soybean refuge area management, and by inference, refuge management strategies in other Bt crops.

## 2. Materials and Methods

### 2.1. Insect Rearing

Susceptible strains of *C. virescens* were obtained from Benzon Research (Carlisle, Pennsylvania). The larvae were reared on tobacco budworm artificial diet purchased from Southland Products (Lake Village, Arkansas). This diet was a dry premix and was prepared following the manufacturer’s suggested protocol. The *C. virescens* adults were fed a 10% honey solution and water via cotton wicks in Petri dishes placed on the bottom of adult cages. All insect stages were maintained in chambers at constant environmental conditions (25 ± 2 °C, 60 ± 10% RH [relative humidity], and 14 h Light: 10 h Dark).

### 2.2. Insecticides

The commercial formulations of diamide insecticides used in the bioassays were Prevathon^®^ 5 SC (chlorantraniliprole, 5% active ingredient [a.i.]) supplied by DuPont and Belt^®^SC (flubendiamide, 39% a.i.) by Bayer CropScience.

### 2.3. Larval Toxicity Bioassay

For determination of sublethal concentrations, eight concentrations of chlorantraniliprole (0.56 to 32.0 ng mL^−1^) and nine concentrations of flubendiamide (1.8 to 180 ng mL^−1^) on a logarithmic scale were tested on newly molted third instar larvae of *C. virescens* using a diet-incorporated bioassay [16]. Insecticide concentrations were dissolved in distilled water to create a stock solution and then serial dilutions of desired concentrations were performed. Forty mL of insecticide solutions at desired concentrations were added to diet to yield 400 mL of diet when it dropped to 55 °C. Control diets were produced with the same procedure using 40 mL of distilled water. Three mL of diet was placed into 30-mL clear plastic cups. Individual third instar *C. virescens* larvae were placed in each cup after the diet cooled. Ninety third instar larvae were tested per treatment. Insect mortality was evaluated daily for seven days following exposure to treated diets. Larvae were considered dead if they did not show head movement or peristaltic contractions when touched with a paintbrush. Moribund larvae were scored as alive [38]. The mortality of the treated insects was corrected using the control treatment mortality according Abbott’s formula [39]. The corrected data were submitted to Probit analysis, using the Polo-Plus program (LeOra Software^®^, Berkeley, CA) to analyze the concentration-mortality relationship [40]. The lethal concentration [LC]_50_, LC_40_, LC_30_ and the corresponding confidence intervals (95% CI) for chlorantraniliprole and flubendiamide were estimated and the values were considered different when there was no overlap of the 95% CI (confidence interval).

### 2.4. Sublethal Effects on C. virescens on Life History Traits and Life Table Parameters

Based on previous research, the LC_30_ was selected as a representative sublethal concentration to assess effects on life history traits and life table parameters of *C. virescens* [30,31]. Newly molted third instar larvae were exposed for 7 days on treated diet. Studies were scheduled in subsets over time, to ensure sufficient numbers of adult survivors (male and female) on different days for tethered flight bioassays (see below). Thus, 1200, 600, and 950 larvae were reared for chlorantraniliprole LC_30_, flubendiamide LC_30_ and for the control treatment, respectively. Surviving larvae were then placed on untreated diet until pupation. New diet was provided as needed. A randomly selected, representative number of surviving larvae in each treatment were weighed after 7 days on the treated diets using an analytical balance. Larvae placed on untreated diet were weighed after another 4 and 11 days.

Larval mortality and development were monitored daily. After pupation, each insect was sexed and weighed. Pupae were individually placed in plastic cups (30 mL) and examined daily for adult emergence. A representative number of newly emerged adults (<24 h) were weighed before they fed. The numbers of normal and deformed adults were recorded. Adults were considered deformed if they were unable to shed the pupal exuvium or had wing deformities. A pair of nondeformed, newly emerged moths (<24 h) was introduced into a rearing cage. The rearing cage was 12-cm-high with top and bottom diameters of 16 and 14 cm, respectively, and contained an inner sheet of paper and a transparent fabric top, both of which served as oviposition substrates. A Petri dish (6 cm diameter) was placed on the cage bottom with a wick of cotton soaked in a 10% honey solution for adult food. Oviposition substrates and food supplies were replaced as needed. In all treatments, at least 24 pairs of moths were used for studies. Eggs laid by each pair were counted daily until female death.

Male and female adult longevity was recorded. Males were not replaced if they died before the female. To evaluate fertility (percentage of eggs that hatched), a minimum of 800 eggs from five random pairs per treatment were collected on the third day of oviposition. Immature development and survival, female oviposition period, fecundity, fertility, and female longevity were used to construct life tables for each treatment. Parameters measured included the intrinsic rate of increase (*r*), the finite rate of increase (*λ*), net reproductive rate (*R_o_*), and mean generation time (*T*).

### 2.5. Sublethal Effects on C. virescens’ Flight Performance

The sublethal effects of chlorantraniliprole and flubendiamide on *C. virescens’* flight performance were estimated using tethered flight on automated flight mill apparatus [41,42,43]. The flight mill consisted of a wooden base with a lightweight, aerodynamic, stainless arm (30 cm length, 0.95 m circumference) with a Teflon rod pivot and magnetic levitation that essentially eliminated friction. Flight rotations were counted with a magnetic sensor (Optec, Inc., Lowell, MI, USA) that was monitored continuously by a computer via a digital input/output board (National Instruments, Austin, TX). The device consisted of 24 flight mills that were run simultaneously. For flight assays, newly emerged (<24 h old) unmated adults of *C. virescens* that were previously exposed to insecticide treatments as detailed above were randomly selected from treatment cages. These moths were separate from those used in life history and life table studies. Only adults with nondeformed wings were assayed. Males and females of *C. virescens* were flown on different days to avoid any disturbance from sex pheromone.

Adults were fed with a 10% honey solution for 4 h before tethering while being maintained in environmental chambers (25 ± 2 °C and 60 ± 10% RH). Moths were anesthetized in a freezer (−15 °C) for 5–7 min. Stainless-steel entomological pins (number 00) with the nylon head removed were used to tether the moths. The cut end was inserted into one end of a computer connector pin that could then be connected to the flight mill arm. A small cork (ca. 2 mm square) attached to the pointed end of the pin served as the tethering point. Gel super glue with bond activator (Loctite^®^; Henkel Corporation, Düsseldorf, Germany) was used to attach the tether to the moth’s prothorax after clearing the scales with a small paintbrush. To minimize wing movement and stress before the flight tests, the moth was placed in a Styrofoam box and positioned, so its legs were in contact with the substrate. Moths were refrigerated (4 °C) for 10 min before the start of the flight assay. Due to variable effects of the insecticides on insect development and the large number of insects that needed to be reared, treatments could not be blocked over time in the flight chamber. To the degree possible, the insects flown on any given day were all either male or female from one insecticide treatment and the control.

The flight mill system was located in an environmentally controlled room (25 ± 2 °C and 60 ± 10% RH) and assays were conducted from 7 p.m. to 7 a.m. during the dark phase of the daily cycle. Each moth was flown a single night. A custom LabView (National Instruments, Austin, TX) computer program automatically recorded data for each of 24 stations including the clock time of the beginning and end of each flight, and the number of revolutions of each flight. These data were then used to calculate flight duration (s), flight distance (m, one rotation = 0.95 m), and flight speed (m/s). The time between flight bouts also was calculated (arrest time in s). Additional calculations were made to estimate total flight time, total distance, and total arrest time over the 12-h assay.

For analyses, *C. virescens’* individual flight durations were categorized as sustained (>30 min) or unsustained (≤30 min). This delineation was consistent with a gap in the distribution of flight durations and has been used by many other researchers to delineate sustained flights in various insect species [43,44,45]. Analyses also examined the timing and duration of the first flight of each moth, again delineated as sustained or unsustained.

### 2.6. Data Analysis

The data were subjected to exploratory analyses to assess the assumptions of normality of residuals [46], homogeneity of variance of treatments and additivity of the model [47]. The flight data were log (duration, distance, speed, and arrest) or square-root transformed (sum of duration, distance, and arrest) as required prior to application of ANOVA. These transformations adequately normalized the residuals. The design was a completely randomized two-factor model with insecticides and sex as fixed effects for flight data and most of the life history data. Data were analyzed using Proc GLIMMIX (generalized linear mixed models) [48], and the SLICEDIFF (simple effects test) options within LSMEANS (least square means) were used to examine simple effects. Mean separation was done using the Tukey option, which controls for experiment-wise error rates. One-way models were used for some life history data where sex was not a factor (e.g., fecundity). Proportional data were analyzed with one-way ANOVA in JMP (SAS Institute, Cary, NC, USA) and the effects were analyzed with a Chi-square test (χ^2^; α = 0.05) [48].

Life table statistics for *C. virescens* were estimated using a matrix model approach in Pop Tools [49]. Matrices for each treatment were parameterized as detailed in Naranjo [50] to estimate λ, the finite growth rate (insects/female/day); *r*, intrinsic rate of increase; *Ro*, net reproductive rate; and *T,* generation time (days) for each treatment. Confidence intervals were estimated by bootstrap resampling with 5000 iterations. Permutation testing was used to compare life table parameters between treatments with a Bonferroni correction for multiple comparisons. The test statistic was simply the difference between treatment parameters and the *p*-value was estimated as the number of times the resampled test statistic exceeded the original test statistic out of 5000 iterations.

## 3. Results

### 3.1. Larval Toxicity Bioassay

*Chloridea virescens’* third instar larvae were more susceptible to the lethal concentrations of chlorantraniliprole than flubendiamide. The LC_50_ of chlorantraniliprole was 4.819 ng mL^−1^ and corresponded to about a 7-fold lower concentration than the flubendiamide LC_50_ (27.972 ng mL^−1^). Likewise, chlorantraniliprole LC_40_ and LC_30_ concentrations (4.007 and 3.289 ng mL^−1^) were approximately 4.5- and 4-fold lower than flubendiamide LC_40_ and LC_30_ concentrations (18.583 and 11.997 ng mL^−1^) (Table 1).

### 3.2. Sublethal Effects on C. virescens’ Life History and Life Table Parameters

Significant differences were found among treatments for larval survival when fed on treated diets for the first 7 days (F-value = 34.83; df (degrees of freedom) = 2, 2547; *p* < 0.0001; Figure 1). *C. virescens’* survival was lowest in the chlorantraniliprole treatment during this period, in comparison with flubendiamide and the control. After placing exposed larvae on untreated diets, significant differences also were observed among treatments, but here insecticide exposure reduced survival compared with the control. (F = 72.17; df = 2, 2034; *p* < 0.0001) (Figure 1).

Both male and female larval development times were longer with exposure to sublethal insecticide doses and the effect was more pronounced with chlorantraniliprole (Table 2). Likewise, pre-pupae development times for both sexes were longer following insecticide exposure. Female pupal stage duration did not differ among treatments, but male pupal duration was longest with exposure to flubendiamide (Table 2). Larval survival was highest in the controls compared with the insecticide treatments (Table 3). However, neither pupal survival nor the sex ratio of resulting adults differed among treatments (Table 3). 

Following 7 days of exposure to insecticides, larval weight was much lower compared with the control (Table 4). For exposed larvae allowed to feed for 4 additional days on untreated diet, those initially on chlorantraniliprole diets weighed less compared with those exposed to flubendiamide (Table 4). Comparisons to the control were not possible because after 11 days the control larvae had already reached the pre-pupal or pupal stage. Pupal weight did not differ between chlorantraniliprole and the control; insects exposed to flubendiamide weighed the least for both sexes (Table 4). A similar pattern was observed for adult weight.

Adult male longevity was highest with exposure to chlorantraniliprole compared with flubendiamide and the control, but female longevity was unaffected (Table 5). Flubendiamide reduced fecundity (total number of eggs/female) compared to chlorantraniliprole or the control. Chlorantraniliprole reduced egg fertility compared to flubendiamide and the control (Table 5). Overall, sublethal doses of chlorantraniliprole and flubendiamide, compared with the control, reduced the finite rate of increase (λ), the innate capacity of population increase (*r*), net reproductive rate (*R_o_*), and the resultant generation time (*T*) (Table 6). Net reproductive rates also differed between the two insecticide exposures.

### 3.3. Sublethal Effects on C. virescens’ Flight Performance

For the duration of continuous individual sustained flights greater than 30 min, differences among treatments were observed for males (F = 3.00; df = 2, 202; *p* = 0.05), but not females (*p* = 0.62) (Figure 2A). Males in the flubendiamide treatments exhibited the longest mean flight duration while the shortest was observed with chlorantraniliprole. The mean flight distance for females ranged from 6451–8151 and males 6664–9311 m. There were no treatments effects on flight distance, flight speed, the rest period between flights (Figure 2B–D), or the number of flights and total flight distance over the 12-h assay period for either sex (Figure 3B,D). In contrast, the mean total flight duration over the 12-h assay period was higher for males in the flubendiamide treatment compared with chlorantraniliprole or the control (F = 3.18; df = 2, 202; *p* = 0.0436; Figure 3A).

For continuous flights less than 30 min, treatments once again only affected male moths. Males in the control treatment had the longest flight durations and those in the chlorantraniliprole treatment had the shortest durations (F = 3.41; df = 2, 481; *p* = 0.0337) (Figure 4A). Control males also flew the longest distance while males exposed to chlorantraniliprole flew the shortest distance (F = 3.83; df = 2, 481; *p* = 0.0224; Figure 4B). The flight speed of males was fastest in the control and slowest in the chlorantraniliprole treatment (F = 5.71; df = 2, 481; *p* = 0.0036; Figure 4C). Rest periods between flights did not differ by treatment (Figure 3D). No differences among treatments were observed in the mean cumulative duration, distance, rest periods, and number of flight attempts over the entire 12-h assay (Figure 5A–D).

For first flights greater than 30 min in duration, no differences were observed in either sex for duration, distance, speed, or rest periods (Figure 6A–D). However, for first flight durations less than 30 min, differences were observed for flight duration (F = 6.31; df = 2, 406; *p* = 0.002; Figure 7A), distance (F = 7.61; df = 2, 406; *p* = 0.0006; Figure 7B), and flight speed for males (F = 8.24; df = 2, 406; *p* = 0.0003; Figure 7C). Males flew about twice as long and twice the distance in the control compared with the insecticide treatments (Figure 7A,B). Males flew the slowest in the flubendiamide treatment (Figure 7C). No differences were observed in male or female rest periods (Figure 7D).

A high percentage of tethered moths flew in the assay system, ranging from 69–95% depending on treatment. The proportion of tethered moths engaged in unsustained flights differed among treatments for females (χ^2^ = 12.15; df = 2, 280; *p* = 0.0023) and males (χ^2^ = 10.56; df = 2, 319; *p* = <0.0001) (Figure 8A). Averaged over both sexes, moths exposed to chlorantraniliprole had the highest propensity for flight (90%) followed by the control (84%) and flubendiamide (66%). There was no difference in the propensity of moths engaged in sustained flight (*p* > 0.10, Figure 8B). Over all treatments, about 35% of moths engaged in sustained flights greater than 30 min in duration.

## 4. Discussion

In Brazil, Bt soybean has been highly effective in controlling *C. virescens* [6,7]. Nevertheless, foliar sprays could be necessary for lepidopteran pest management in refuge areas [14,15,27]. Such sprays would reduce the abundance of susceptible moths through direct mortality, but there may be additional sublethal effects on surviving insects as the concentrations of insecticides degrade after application [25,26]. Models of the high-dose, refuge strategy assume at least 500 susceptible adults will emerge for every resistant moth, then randomly mate with rare and resistant insects (RR) from the Bt fields, resulting in susceptible heterozygous progeny (RS) [9,51,52]. While these models account for direct mortality of the target pest through the size and placement of the refuge (e.g., 20% of Bt crop), it is less clear if they also account for the more subtle biological and behavioral effect that could impact survivors exposed to sublethal insecticide doses [53,54]. Here, we demonstrated that flubendiamide and chlorantraniliprole, two diamide insecticides that are popular insect control choices for growers in Brazil [55,56], are toxic to *C. virescens* larvae. In addition, sublethal concentrations of these insecticides at a dose represented by the LC_30_, were associated with changes in life history traits that ultimately affected population growth characteristics and phenology. Additional effects were observed in the flight behavior of male moths.

These life history trait effects included reductions in larval, pupal, and adult weight; prolongation of larval and pupal development times; reductions in adult male longevity; and reductions in female fertility and fecundity. There also were additional reductions in larval survival, which is usual in studies of this nature, but suggests that our sublethal doses may have been slightly too high. Overall, our results are consistent with other studies on diamide sublethal effects [29,30,31,57,58]. Nonetheless, the insect suffered multiple biological changes as a result of exposure to what would represent degraded insecticide concentrations in the field [59,60]. The biological traits affected are interrelated and so each effect does not represent an independent outcome of sublethal stress. Life table statistics allowed us to integrate these observed effects into more meaningful metrics. We observed reductions in net reproductive rates and intrinsic rates of increase, and prolonged generation times due to sublethal exposure. These changes could potentially lead to asynchronous target pest population growth and phenology between Bt and non-Bt soybeans, and, ultimately, reduce the probability of random matings between susceptible and putatively resistant insects.

Several additional factors need to be considered in determining whether and to what degree these sublethal effects will disrupt resistance management. For example, diamides insecticides have been shown to have low toxicity to natural enemies [18,61]. Thus, target pests in refuge crops could be subject to higher levels of predation and parasitism from conserved natural enemies. This additional mortality may further reduce the number of susceptible moths generated in refuge fields and the effect could be further enhanced by the increased susceptibility of sublethally affected prey to natural enemy attack [62]. In contrast, there may be fitness costs associated with resistance to Bt proteins in the target pest that could affect life history traits in much the same way as the sublethal effects observed here. This might negate the impacts associated with asynchronicity in population growth and phenology previously discussed [63]. However, the transgenic soybean events used in Brazil express a high dose of the Cry 1Ac, and studies suggest that *C. virescens’* larvae from first through fifth instar are highly susceptible to Cry 1Ac, leading to 100% mortality [6,7]. Thus, it is unclear if fitness costs would be relevant given essentially no survivors in Bt soybean at the present time. Finally, natural enemy activity in the Bt crop might also contribute to delays in resistance evolution [64,65]. Overall, there are many interacting factors influencing the outcome of the high-dose, refuge strategy that may need to be more carefully studied and modelled.

Given the need for both sufficient numbers of susceptible moths and populations synchronized to produce adults during the same time period in Bt and non-Bt fields, the dispersal ability of susceptible moths should be sufficient to ensure cross mating of susceptible and resistant moths [9,51,52,66]. IRAC technical recommendations in Brazil recommend at least a 20% structured refuge area within a maximum distance of 800 m (0.5 mile) from Bt soybeans [15,67]. We found that sublethal flubendiamide exposure appeared to reduce the flight propensity of moths while chlorantraniliprole increased flight propensity relative to the control. These differences were apparent only for unsustained flights <30 min in duration; the propensity for sustained flight was unchanged by treatment. For insects that flew, we observed effects of sublethal exposure on several aspects of flight behavior, but only for male moths. Why female moth flight was unaffected is not clear. One hypothesis is that there was a trade-off between flight capacity and fecundity, which was reduced with sublethal exposure [68]. Insect flight and reproduction are critical ecological processes that are both energetically costly [69,70]. Flight fuel production can compete with energy used for ovarian development and oogenesis [69,70,71]. *C. virescens* females may have invested in flight energy at the expense of reproduction, but additional work will be needed to test this hypothesis.

Sublethal concentrations of flubendiamide reduced pupal and adult weight, in addition to fecundity. Studies have shown that the binding of flubendiamide on *C. virescens’* thoracic muscles was four times higher than cyantraniliprole, another anthranilic similar to chlorantraniliprole [72]. Therefore, we hypothesized that the absence of negative impact of chlorantraniliprole LC_30_ on *C. virescens* fecundity might be associated with differential ryanodine binding on the RyR complex [72,73]. This differential effect of the two insecticides may also help explain the lower flight propensity observed for moths exposed to flubendiamide. It should be noted that we tested only newly emerged and unmated moths. These individuals may use their lipid energy sources for flights rather than for egg maturation [68,74], corroborating the lack of sublethal effects’ exposure on flight capacity. It is also possible that females have more efficient detoxification systems for metabolizing insecticides [75]. Thus, the muscle-related effects of diamides may have only affected males [24]. Future research is needed to test these hypotheses on differential female and male effects.

Flight mills have been widely used to measure flight behavior in a wide diversity of insects and to examine the effects of a wide array of biological and ecological factors [32,33,34,35,36,43,44,45]. Still, the approach represents a highly artificial system that makes it difficult to tie behavior in the laboratory to that in the field [43,76,77,78]. The best approach is to use flight mill results in a comparative fashion [43] and/or to think of flight mill results as a measure of the biological potential of a species [79,80,81]. Here, despite the negative impact of sublethal exposure on some flight performance aspects of *C. virescens* males, both sexes flew approximately 1.7 to 2.7 times the maximum distance (800 m) required between Bt soybean fields and refuge areas when engaged in unsustained flight in a single 12-h assay period. This figure jumped to 10- to 14-fold for moths engaged in sustained flight, although only about a third of moths do so. Thus, both sexes have the potential to more than adequately move the required 800 m. Still, fewer moths exposed to flubendiamide would be expected to engage in unsustained flight, while those exposed to chlorantraniliprole seemed to be more active than the control. The overall impact of these differences is difficult to judge in the field, but flubendiamide use could reduce the number of moths traveling the required 800 m. Previous work has demonstrated that *C. virescens* adults can fly more than 7.5 km in their lifetimes [1,3,60,82], although the moth engages in facultative migration behavior that is modified by the environment and other biological factors [76,77,78]. Only the mean distance of individual male flights less than 30 min were reduced by sublethal exposure. Mean distance of flight over 30 min and cumulative flight distance over the 12-h assay did not differ between insecticide treatments or sex. Thus, if we assume that unexposed moths are capable of reaching the refuge fields (based on the refuge distance requirement), then there is no indication that sublethally affected moths could not do the same, albeit the total number doing this could be reduced with use of flubendiamide.

## 5. Conclusions

Overall, we found that sublethal exposure of *C. virescens* to several common diamide insecticides can reduce population growth and alter phenological timing. This could potentially disrupt random mating between susceptible moths from the refuge and resistant moths from Bt fields and interfere with resistance mitigation. The extent of the impact would likely depend on the actual sublethal dose the insects would encounter and this would certainly change over time [25,26,59,60,83,84]. As noted, these effects also could be blunted by fitness-related costs in resistant moths. Our flight mill results suggest that this sublethal exposure would not be expected to reduce the moths’ ability to disperse an adequate distance from non-Bt refuges, but additional research will be needed to define the extent of sublethal exposure in the field. Our results may be applicable to other Bt crops in which *C. virescens* is an economically important pest, such as Bt cotton in Mexico, Puerto Rico, Colombia, and Brazil [85], and to situations where diamides insecticides are used in refuge areas of other crops for tobacco budworm population control. It is likely that the life history effects demonstrated here are not unique to diamide insecticides or even the specific target species and crop examined. Instead, this may be a concern for any effective insecticide used to manage target pests in refuge crops and further work may be needed to examine the underlying assumptions of the high-dose refuge strategy, especially with reference to the number of susceptible moths required to be generated for effective resistance management.

## Figures and Tables

**Figure 1 insects-11-00269-f001:**
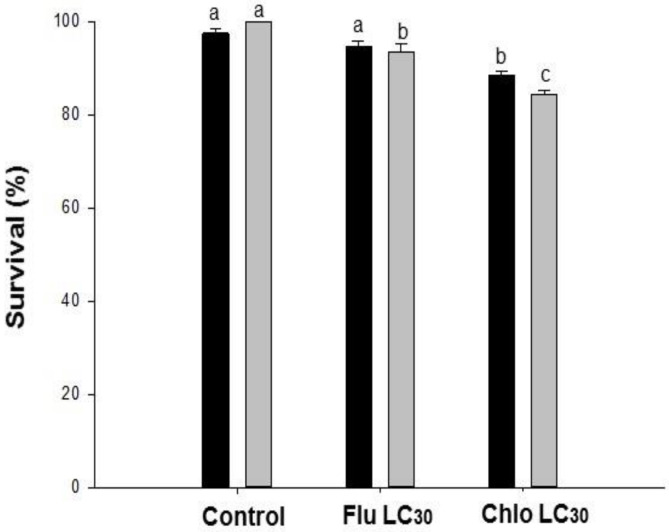
Survival (%) of *Chloridea virescens* larvae (mean ± SE) when exposed for 7 days on treated diet (black bars) with sublethal concentrations of flubendiamide (Flu) and chlorantraniliprole (Chlo), and after feeding on untreated diet four additional days (grey bars). The corresponding sample sizes were: Control (950, 775), flubendiamide (390, 296), and chlorantraniliprole (1210, 775). Mean survival with different letters within the same color of bars are significantly different (Tukey test, *p* < 0.05).

**Figure 2 insects-11-00269-f002:**
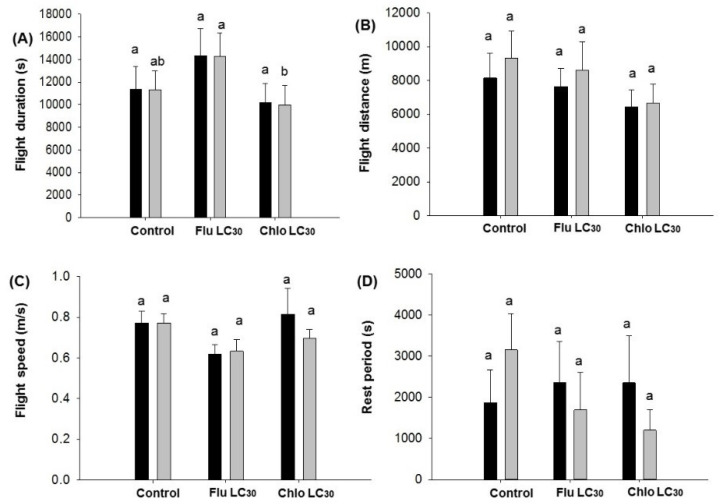
*Chloridea virescens* sustained flights greater than 30 min: (**A**) Mean flight duration, (**B**) mean flight distance, (**C**) mean flight speed, (**D**) mean rest period. The black bars represent female and grey bars represent male moths. Different letters above the standard error bars indicate significant differences based on Tukey–Kramer test (*p* < 0.05).

**Figure 3 insects-11-00269-f003:**
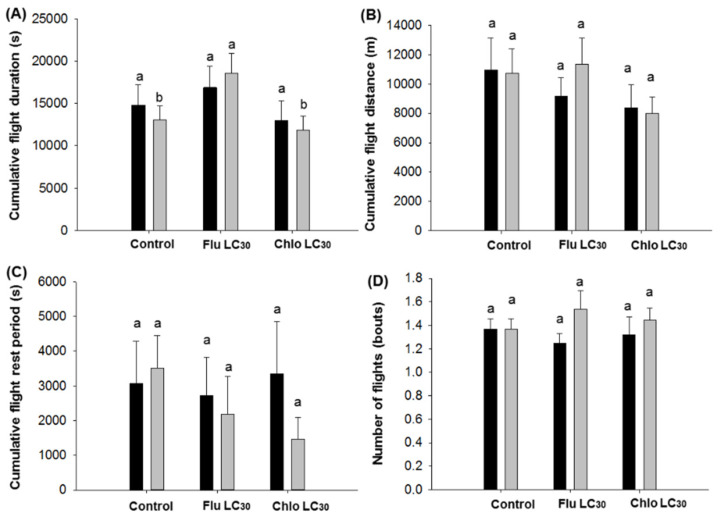
*Chloridea virescens* sustained flights greater than 30 min: (**A**) Cumulative flight duration, (**B**) cumulative flight distance, (**C**) cumulative flight rest period, and (**D**) number of flights (bouts). Black bars represent female and grey bars represent male moths. Different letters above the standard error bars indicate significant differences based on Tukey–Kramer test (*p* < 0.05).

**Figure 4 insects-11-00269-f004:**
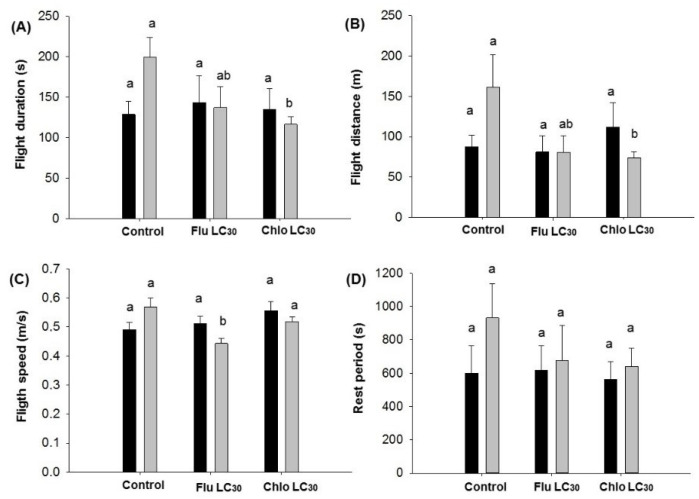
*Chloridea virescens* unsustained flights less than 30 min: (**A**) Mean flight duration, (**B**) mean flight distance, (**C**) mean flight speed, and (**D**) mean rest period. Black bars represent female and grey bars represent male moths. Different letters above the standard error bars indicate significant differences based on Tukey–Kramer test (*p* < 0.05).

**Figure 5 insects-11-00269-f005:**
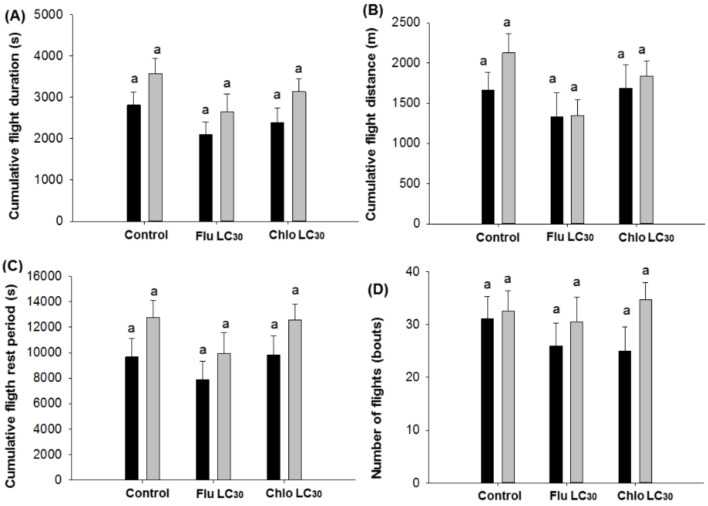
*Chloridea virescens* unsustained flights less than 30 min: (**A**) Cumulative flight duration, (**B**) cumulative flight distance, (**C**) cumulative flight rest period, and (**D**) number of flights (bouts). Black bars represent female and grey bars represent male moths. Different letters above the standard error bars indicate significant differences based on Tukey–Kramer test (*p* < 0.05).

**Figure 6 insects-11-00269-f006:**
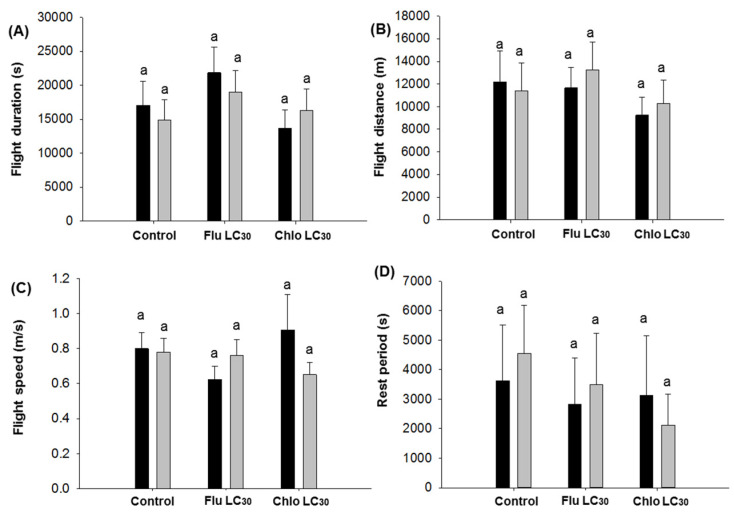
First flight taken by *Chloridea virescens* greater than 30 min: (**A**) Mean flight duration, (**B**) mean flight distance, (**C**) mean flight speed, and (**D**) mean rest time. Black bars represent female and grey bars represent male moths. Different letters above the standard error bars indicate significant differences based on Tukey–Kramer test (*p* < 0.05).

**Figure 7 insects-11-00269-f007:**
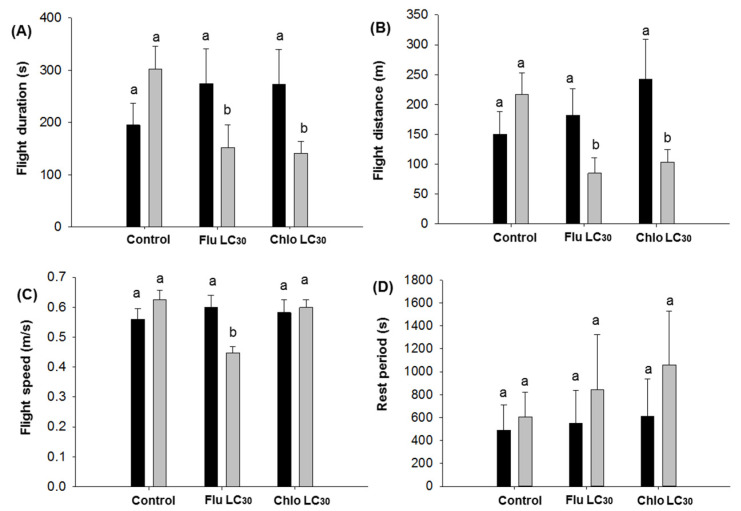
First flights of *Chloridea virescens* less than 30 min: (**A**) Flight duration, (**B**) flight distance, (**C**) flight speed, and (**D**) rest period. Black bars represent female and grey bars represent male moths. Different letters above the standard error bars indicate significant differences based on Tukey–Kramer test (*p* < 0.05).

**Figure 8 insects-11-00269-f008:**
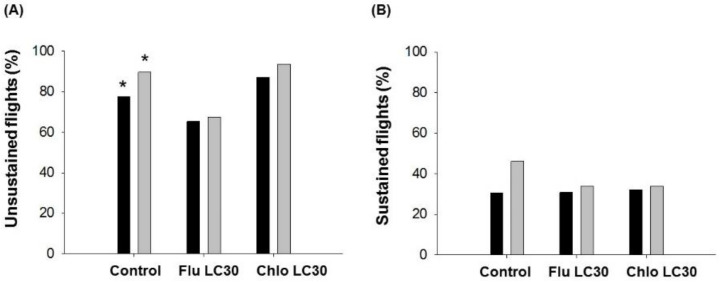
Proportion of *Chloridea virescens* that (**A**) engaged in unsustained (<30 min) or (**B**) sustained flights (>30 min). Black bars represent female and grey bars represent male moths. The asterisks indicate significant differences among treatments for a given comparison (χ^2^, α = 0.05).

**Table 1 insects-11-00269-t001:** Toxicity of flubendiamide and chlorantraniliprole to third instar larvae of *Chloridea virescens* in diet bioassays.

Insecticides	n	Slope (SE) ^a^	LC_50_ (ng mL^−1^) ^b^	LC_40_ (ng mL^−1^)	LC_30_ (ng mL^−1^)	χ^2^ (df) ^c^	*p*
Flubendiamide	900	1.43 (0.10)	27.972 (18.08–39.64)	18.583 (11.71–25.09)	11.997 (8.562–16.014)	11.064 (6)	0.09
Chlorantraniliprole	810	3.16 (0.20)	4.819 (4.02–5.79)	4.007 (3.302–4.786)	3.289 (2.645–3.943)	12.194 (6)	0.06

^a^ Standard error, ^b^ 95% confidence limits, ^c^ Chi-square value (χ^2^) and degrees of freedom (df) as calculated by probit analysis.

**Table 2 insects-11-00269-t002:** Sublethal effects of flubendiamide (Flu) and chlorantraniliprole (Chlo) on larval, pre-pupal, and pupal development time of *Chloridea virescens* under controlled laboratory environmental conditions (25 ± 2 °C, 60 ± 10% RH, and 14L: 10D).

Treatments	Larval Stage (Days)		Pre-Pupal Stage (Days)		Pupal Stage (Days)	
	Female	Male	Female	Male	Female	Male
Control	8.72 ± 0.09 c	8.85 ± 0.08 c	2.02 ± 0.07 b	1.95 ± 0.07 b	11.44 ± 0.03 a	12.81 ± 0.03 b
Flu LC_30_	12.56 ± 0.16 b	12.53 ± 0.13 b	2.54 ± 0.13 a	2.67 ± 0.11 a	11.58 ± 0.07 a	13.02 ± 0.06 a
Chlo LC_30_	14.60 ± 0.083 a	14.71 ± 0.08 a	2.75 ± 0.07 a	2.63 ± 0.07a	11.55 ± 0.03 a	12.81 ± 0.03 b
F	1090.65	1208.32	23.98	26.880	2.820	5.440
df	2, 1835	2, 1835	2, 1835	2, 1835	2, 1734	2, 1734
*p*	<0.0001	<0.0001	<0.0001	<0.0001	0.06	0.004

All values are means ± standard error (SE). Means in same column followed by different letters are significantly different (Tukey test, *p* < 0.05).

**Table 3 insects-11-00269-t003:** Sublethal effects of flubendiamide (Flu) and chlorantraniliprole (Chlo) on larvae and pupal survival and the sex ratio of resulting adult *Chloridea virescens* under controlled laboratory environmental conditions (25 ± 2 °C, 60 ± 10% RH, and 14L: 10D).

Treatments	(%) Larval Survival ^1^	(%) Pupal Survival ^2^		(%) Sex Ratio
		Female	Male	
Control	81.47 ± 1.42 a	93.58 ± 1.12 a	93.50 ± 1.11 a	0.48 a
Flu LC_30_	71.02 ± 2.22 b	93.96 ± 2.02 a	92.50 ± 1.76 a	0.42 a
Chlo LC_30_	67.43 ± 1.26 b	96.67 ± 1.10 a	96.70 ± 1.08 a	0.48 a
F	27.8578	2.0812	3.0694	1.7601
df	2, 2547	2, 878	2, 982	2, 1863
*p*	<0.0001	0.1254	0.0469	0.1723

^1^ Larvae that became pupae, ^2^ pupae that became adults. All values are means ± standard error (SE). Means in same column followed by different letters are significantly different (Tukey test, *p* < 0.05).

**Table 4 insects-11-00269-t004:** Sublethal effects of flubendiamide (Flu) and chlorantraniliprole (Chlo) on larval weight after 7 days of exposure to treated diets, after 4 additional days on untreated diet (11 days total), and on pupal and adult weight of *Chloridea virescens* under controlled laboratory environmental conditions (25 ± 2 °C, 60 ± 10% RH, and 14L: 10D).

Treatments	Larval Weight (g)		Pupal Weight (g)		Adult Weight (g)	
	7 Days	11 Days ^a^	Female	Male	Female	Male
Control	0.342 ± 0.01 a	*	0.268 ± 0.01 a	0.279 ± 0.01 a	0.152 ± 0.01 a	0.142 ± 0.01 a
Flu LC_30_	0.086 ± 0.01b	0.243 ± 0.01 a	0.246 ± 0.01 b	0.247 ± 0.01 b	0.141 ± 0.03 b	0.124 ± 0.01 b
Chlo LC_30_	0.026 ± 0.001 c	0.171 ± 0.01 b	0.276 ± 0.01 a	0.282 ± 0.01 a	0.156 ± 0.01 a	0.145 ± 0.01 a
F	1073.38	23.4	20.27	40.63	8.77	20.340
df	2, 578	1, 329	2, 727	2, 727	2, 396	2, 396
*p*	<0.0001	<0.0001	<0.0001	<0.0001	0.0002	<0.0001

^a^ Larval survival after 7 days on treated diet and 4 additional days on untreated diet. * Comparable weight was not possible because larvae on untreated diets completed development before 11 days. All values are means ± standard error (SE). Means in same column followed by different letters are significantly different (Tukey test, *p* < 0.05 for 3 mean comparisons).

**Table 5 insects-11-00269-t005:** Sublethal effects of flubendiamide (Flu) and chlorantraniliprole (Chlo) on adult longevity, fecundity, and fertility of *Chloridea virescens* under controlled laboratory environmental conditions (25 ± 2 °C, 60 ± 10% RH, and 14L: 10D).

Treatments	Longevity (Days)		Fecundity ^1^	Fertility ^2^ (%)
	Female	Male		
Control	10.00 ± 0.93 a	10.83 ± 0.93 b	1289.45 ± 80.22 a	87.37 ± 1.49 a
Flu LC_30_	9.73 ± 0.89 a	10.11 ± 0.89 b	873.54 ± 80.22 b	83.78 ± 1.20 ab
Chlo LC_30_	11.00 ± 0.85 a	13.38 ± 0.85 a	1193.75 ±72.98 a	81.63 ± 1.27 b
F	0.59	3.88	7.46	4.2987
df	2, 152	2, 152	2, 74	2, 2357
*p*	0.5553	0.023	0.0011	0.0137

^1^ Total number of eggs laid per female, ^2^ percent of eggs hatching. All values are means ± standard error (SE). Means in same column followed by different letters are significantly different (Tukey test, *p* < 0.05).

**Table 6 insects-11-00269-t006:** Life table parameters for *C. virescens* (mean, 95% CI) exposed to sublethal concentrations of flubendiamide (Flu) and chlorantraniliprole (Chlo) under controlled laboratory environmental conditions (25 ± 2 °C, 60 ± 10% RH, and 14L: 10D).

		Treatments	
Life Table Parameters	Control	Flu LC_30_	Chlo LC_30_
*λ* ^a^	1.312 (1.301–1.323) a	1.237 (1.219–1.256) b	1.257 (1.247–1.267) b
*R* ^b^	0.272 (0.263–0.280) a	0.213 (0.198–0.228) b	0.229 (0.221–0.236) b
*R_o_* ^c^	422.7 (377.8–464.1) a	213.5 (167.9–263.6) b	313.2 (285.4–340.8) c
*T* ^d^	37.5 (36.7–38.4) a	40.9 (39.1–42.8) b	43.2 (41.7–44.7) b

^a^ λ, finite growth rate (insects/female/day); ^b^
*r*, intrinsic rate of increase; ^c^
*Ro*, net reproductive rate; ^d^
*T*, generation time (days). Confidence intervals estimated by bootstrap analysis and mean comparisons done with permutation testing (5000 repetitions) corrected for multiple comparisons. Means followed by different letters within a row are significantly different (*p* < 0.0167).

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
