# Peer review of "Sublethal Effects of Diamide Insecticides on Development and Flight Performance of Chloridea virescens (Lepidoptera: Noctuidae): Implications for Bt Soybean Refuge Area Management"

_insects, 2020, doi:10.3390/insects11050269_

Round 1

Reviewer 1 Report

The aim of the study was to evaluate the effects of two insecticides from the diamide family, chlorantraniliprole and flubendiamide on some life history traits of Chloridea virescens and on flight performance. The study is of interest because this pest insect is of economic importance and the two molecules are currently in use in Bt soybean refuge, so understand their effects is essential. I have few concerns regarding the conclusions for the real implication of these results for pest management in Bt refuge, without knowledge of the same life history traits of Bt resistant population of C. virescens it is difficult to assume that insects will be asynchron.

The experiments are properly done and I have only minor comments, please see below:

Results

P5 line 215 and P5 lines 218-219, give explanation about what mean F, dF, no description of these terms have been given in Materials and Methods.

The number of tables and figures is relatively high (6 tables and 8 figures), some of them can be combined. This is more a suggestion than a real criticism.

Minor points :

P4 line 169, please remplace 1900 to 0700 h by 7 pm to 7 am during the dark phase of the daily cycle.

Reviewer 2 Report

Overall, this paper is well-written, and the experimental procedures, results and conclusions are presented in a logical fashion. The paper presents some interesting information and I believe it will be valuable information for the scientific community and should be published. I have no complaints about paper’s presentation. I have only noted a few grammatical edits, which are given below.

Line 45: delete “a” after “express”

Line 57: change “population” to “populations”

Line 69: change “directly” to “direct”

Lines 81-82: change “C. virescens life history traits” to “life history traits of C. virescens

Line 87: change “Chloridia virescens susceptible strains” to “Susceptible strains of C. virescens

Line 88: no need to capitalize “Tobacco Budworm”

Line 90: delete “with”

Line 109: Please explain how mortalities of the treated insects were corrected with mortalities of control insects. Did you use “Abbot’s formula”?

Line 127: change “Adult” to “Adults”

Line 157: “in to” to “into”

Line 227: delete “then”

Line 275: change “comparing to control” to “compared to the control”

Figures 2-6: change “Different letter above the standard error bar” to “Different letters above standard error bars”

Line 347: add space between the period and “Nevertheless”

Lines 351-352: “adults for every resistant moth, that may then randomly mate…” to “adults will emerge for every resistant moth, then randomly mate…”

Line 391: delete “In addition, to” and begin the sentence with “The”

Line 391: delete “produced”

Line 415: change “system” to “systems”

Line 429: change “cyantraniliprole” to “chlorantraniliprole”

Lines 441-444: Move this sentence that begins with “Our results may be….” to line 452 and put it after the sentence that ends with “field” – this makes the conclusions flow better.
